# Distance learning in higher education during COVID-19: The role of basic psychological needs and intrinsic motivation for persistence and procrastination–a multi-country study

Elisabeth R. Pelikan[1]*, Selma Korlat[1], Julia Reiter[1], Julia Holzer[1], Martin Mayerhofer[2], Barbara Schober[1], Christiane Spiel[1], Oriola Hamzallari[3], Ana Uka[4], Jiarui Chen[5], Maritta Välimäki[5,6], Zrinka Puharić[7], Kelechi Evans Anusionwu[8], Angela Nkem Okocha[8], Anastassia Zabrodskaja[8], Katariina Salmela-Aro[9], Udo Käser[10], Anja Schultze-Krumbholz[11], Sebastian Wachs[12], Finnur Friðriksson[13], Hermína Gunnþórsdóttir[13], Yvonne Höller[13], Ikuko Aoyama[14], Akihiko Ieshima[15], Yuichi Toda[16], Jon Konjufca[17], Njomza Llullaku[18], Reda Gedutienė[19], Glorianne Borg Axisa[20], Irena Avirovic Bundalevska[21], Angelka Keskinova[21], Makedonka Radulovic[21], Aleksandra Lewandowska-Walter[22], Justyna Michałek-Kwiecień[22], Piotr Plichta[23], Jacek Pyżalski[24], Natalia Walter[24], Cristina Cautisanu[25], Ana Iolanda Voda[26], Shang Gao[27], Sirajul Islam[27], Kai Wistrand[27], Michelle F. Wright[28], Marko Lüftenegger[1,29]

1 Department of Developmental and Educational Psychology, Faculty of Psychology, University of Vienna, Vienna, Austria, 2 Department of Mathematics, Faculty of Mathematics, University of Vienna, Vienna, Austria, 3 Department of Psychology, Faculty of Education, Aleksandër Moisiu University, Durrës, Albania, 4 Department of Educational Sciences, Faculty of Philology and Education, Bedër University, Tirana, Albania, 5 Xiangya School of Nursing, Central South University, Changsha, China, 6 Department of Nursing Science, University of Turku, Turku, Finland, 7 Study of Nursing, University of Applied Sciences Bjelovar, Bjelovar, Croatia, 8 Baltic Film, Media and Arts School, Tallinn University, Tallinn, Estonia, 9 Faculty of Educational Sciences, University of Helsinki, Helsinki, Finland, 10 Department of Psychology, University of Bonn, Bonn, Germany, 11 Chair of Educational Psychology, Technische Universität Berlin, Berlin, Germany, 12 Department of Educational Studies, University of Potsdam, Potsdam, Germany, 13 Faculty of Education, University of Akureyri, Akureyri, Iceland, 14 Department of Global Education, Tsuru University, Tsuru, Japan, 15 Career Center, Osaka University, Osaka University, Suita, Japan, 16 Graduate School of Education, Osaka Kyoiku University, Kashiwara, Japan, 17 Department of Psychology, Faculty of Philosophy, University of Prishtina 'Hasan Prishtina', Pristina, Kosovo, 18 Department of Social Work, Faculty of Philosophy, University of Pristina 'Hasan Prishtina', Pristina, Kosovo, 19 Department of Psychology, Faculty of Social Sciences and Humanities, Klaipėda University, Klaipėda, Lithuania, 20 Geography Department, Junior College, University of Malta, Msida, Malta, 21 Institute of Family Studies, Faculty of Philosophy, Ss. Cyril and Methodius University in Skopje, Skopje, North Macedonia, 22 Institute of Psychology, Faculty of Social Science, University of Gdańsk, Gdańsk, Poland, 23 Faculty of Historical and Pedagogical Sciences, University of Wrocław, Wrocław, Poland, 24 Faculty of Educational Studies, Adam Mickiewicz University, Poznań, Poland, 25 CERNESIM Environmental Research Center, Alexandru Ioan Cuza University, Iaşi, Româna, 26 Social Sciences and Humanities Research Department, Institute for Interdisciplinary Research, Alexandru Ioan Cuza University of Iaşi, Iaşi, Româna, 27 Department of Informatics, Örebro University School of Business, Örebro University, Örebro, Sweden, 28 Faculty of Social Studies, Penn State University, State College, Pennsylvania, United States of America, 29 Department for Teacher Education, Centre for Teacher Education, University of Vienna, Vienna, Austria

* elisabeth.pelikan@univie.ac.at



**Data Availability Statement:** Data is now publicly available: Pelikan ER, Korlat S, Reiter J, Lüftenegger M. Distance Learning in Higher

## Abstract

Due to the COVID-19 pandemic, higher educational institutions worldwide switched to emergency distance learning in early 2020. The less structured environment of distance learning

Education During COVID-19: Basic Psychological Needs and Intrinsic Motivation 2021. doi:10. 17605/OSF.IO/8CZX3.

**Funding:** This work was funded by the Vienna Science and Technology Fund (WWTF) [https://www.wwtf.at/] and the MEGA Bildungsstiftung [https://www.megabildung.at/] through project COV20-025, as well as the Academy of Finland [https://www.aka.fi] through project 308351, 336138, and 345117. BS is the grant recipient of COV20-025. KSA is the grant recipient of 308351, 336138, and 345117. Open access funding was provided by University of Vienna. The funders had no role in study design, data collection and analysis, decision to publish, or preparation of the manuscript.

**Competing interests:** The authors have declared that no competing interests exist.

forced students to regulate their learning and motivation more independently. According to self-determination theory (SDT), satisfaction of the three basic psychological needs for autonomy, competence and social relatedness affects intrinsic motivation, which in turn relates to more active or passive learning behavior. As the social context plays a major role for basic need satisfaction, distance learning may impair basic need satisfaction and thus intrinsic motivation and learning behavior. The aim of this study was to investigate the relationship between basic need satisfaction and procrastination and persistence in the context of emergency distance learning during the COVID-19 pandemic in a cross-sectional study. We also investigated the mediating role of intrinsic motivation in this relationship. Furthermore, to test the universal importance of SDT for intrinsic motivation and learning behavior under these circumstances in different countries, we collected data in Europe, Asia and North America. A total of $N = 15,462$ participants from Albania, Austria, China, Croatia, Estonia, Finland, Germany, Iceland, Japan, Kosovo, Lithuania, Poland, Malta, North Macedonia, Romania, Sweden, and the US answered questions regarding perceived competence, autonomy, social relatedness, intrinsic motivation, procrastination, persistence, and socio-demographic background. Our results support SDT's claim of universality regarding the relation between basic psychological need fulfilment, intrinsic motivation, procrastination, and persistence. However, whereas perceived competence had the highest direct effect on procrastination and persistence, social relatedness was mainly influential via intrinsic motivation.

## Introduction

In early 2020, countries across the world faced rising COVID-19 infection rates, and various physical and social distancing measures to contain the spread of the virus were adopted, including curfews and closures of businesses, schools, and universities. By the end of April 2020, roughly 1.3 billion learners were affected by the closure of educational institutions [1]. At universities, instruction was urgently switched to distance learning, bearing challenges for all actors involved, particularly for students [2]. Moreover, since distance teaching requires ample preparation time and situation-specific didactic adaptation to be successful, previously established concepts for and research findings on distance learning cannot be applied undifferentiated to the emergency distance learning situation at hand [3].

Generally, it has been shown that the less structured learning environment in distance learning requires students to regulate their learning and motivation more independently [4]. In distance learning in particular, high intrinsic motivation has proven to be decisive for learning success, whereas low intrinsic motivation may lead to maladaptive behavior like procrastination (delaying an intended course of action despite negative consequences) [5, 6]. According to self-determination theory (SDT), satisfaction of the three basic psychological needs for autonomy, competence and social relatedness leads to higher intrinsic motivation [7], which in turn promotes adaptive patterns of learning behavior. On the other hand, dissatisfaction of these basic psychological needs can detrimentally affect intrinsic motivation. According to SDT, satisfaction of the basic psychological needs occurs in interaction with the social environment. The context in which learning takes place as well as the support of social interactions it encompasses play a major role for basic need satisfaction [7, 8]. Distance learning, particularly when it occurs simultaneously with other physical and social distancing

measures, may impair basic need satisfaction and, in consequence, intrinsic motivation and learning behavior.

The aim of this study was to investigate the relationship between basic need satisfaction and two important learning behaviors—procrastination (as a consequence of low or absent intrinsic motivation) and persistence (as the volitional implementation of motivation)—in the context of emergency distance learning during the COVID-19 pandemic. In line with SDT [7] and previous studies (e.g., [9]), we also investigated the mediating role of intrinsic motivation in this relationship. Furthermore, to test the universal importance of SDT for intrinsic motivation and learning behavior under these specific circumstances, we collected data in 17 countries in Europe, Asia, and North America.

## The fundamental role of basic psychological needs for intrinsic motivation and learning behavior

SDT [7] provides a broad framework for understanding human motivation, proposing that the three basic psychological needs for autonomy, competence, and social relatedness must be satisfied for optimal functioning and intrinsic motivation. The need for autonomy refers to an internal perceived locus of control and a sense of agency. In an academic context, students who learn autonomously feel that they have an active choice in shaping their learning process. The need for competence refers to the feeling of being effective in one's actions. In addition, students who perceive themselves as competent feel that they can successfully meet challenges and accomplish the tasks they are given. Finally, the need for social relatedness refers to feeling connected to and accepted by others. SDT proposes that the satisfaction of each of these three basic needs uniquely contributes to intrinsic motivation, a claim that has been proved in numerous studies and in various learning contexts. For example, Martinek and colleagues [10] found that autonomy satisfaction was positively whereas autonomy frustration was negatively related to intrinsic motivation in a sample of university students during COVID-19. The same held true for competence satisfaction and dissatisfaction. A recent study compared secondary school students who perceived themselves as highly competent in dealing with their school-related tasks during pandemic-induced distance learning to those who perceived themselves as low in competence [11]. Students with high perceived competence not only reported higher intrinsic motivation but also implemented more self-regulated learning strategies (such as goal setting, planning, time management and metacognitive strategies) and procrastinated less than students who perceived themselves as low in competence. Of the three basic psychological needs, the findings on the influence of social relatedness on intrinsic motivation have been most ambiguous. While in some studies, social relatedness enhanced intrinsic motivation (e.g., [12]), others could not establish a clear connection (e.g., [13]).

Intrinsic motivation, in turn, is regarded as particularly important for learning behavior and success (e.g., [6, 14]). For example, students with higher intrinsic motivation tend to engage more in learning activities [9, 15], show higher persistence [16] and procrastinate less [6, 17, 18]. Notably, intrinsic motivation is considered to be particularly important in distance learning, where students have to regulate their learning themselves. Distance-learning students not only have to consciously decide to engage in learning behavior but also persist despite manifold distractions and less external regulation [4].

Previous research also indicates that the satisfaction of each basic need uniquely contributes to the regulation of learning behavior [19]. Indeed, studies have shown a positive relationship between persistence and the three basic needs (autonomy [20]; competence [21]; social relatedness [22]). Furthermore, all three basic psychological needs have been found to be related to procrastination. In previous research with undergraduate students, autonomy-supportive

teaching behavior was positively related to satisfaction of the needs for autonomy and competence, both of which led to less procrastination [23]. A qualitative study by Klingsieck and colleagues [18] supports the findings of previous studies on the relations of perceived competence and autonomy with procrastination, but additionally suggests a lack of social relatedness as a contributing factor to procrastination. Haghbin and colleagues [24] likewise found that people with low perceived competence avoided challenging tasks and procrastinated.

SDT has been applied in research across various contexts, including work (e.g., [25]), health (e.g., [26]), everyday life (e.g., [27]) and education (e.g., [15, 28]). Moreover, the pivotal role of the three basic psychological needs for learning outcomes and functioning has been shown across multiple countries, including collectivistic as well as individualistic cultures (e.g., [29, 30]), leading to the conclusion that satisfaction of the three basic needs is a fundamental and universal determinant of human motivation and consequently learning success [31].

## Self-determination theory in a distance learning setting during COVID-19

As Chen and Jang [28] observed, SDT lends itself particularly well to investigating distance learning, as the three basic needs for autonomy, competence and social relatedness all relate to important aspects of distance learning. For example, distance learning usually offers students greater freedom in deciding where and when they want to learn [32]. This may provide students with a sense of agency over their learning, leading to increased perceived autonomy. At the same time, it requires students to regulate their motivation and learning more independently [4]. In the unique context of distance learning during COVID-19, it should be noted that students could not choose whether and to what extent to engage in distance learning, but had to comply with external stipulations, which in turn may have had a negative effect on perceived autonomy. Furthermore, distance learning may also influence perceived competence, as this is in part developed by receiving explicit or implicit feedback from teachers and peers [33]. Implicit feedback in particular may be harder to receive in a distance learning setting, where informal discussions and social cues are largely absent. The lack of face-to-face contact may also impede social relatedness between students and their peers as well as students and their teachers. Well-established communication practices are crucial for distance learning success (see [34] for an overview). However, providing a nurturing social context requires additional effort and guidance from teachers, which in turn necessitates sufficient skills and preparation on their part [34, 35]. Moreover, the sudden switch to distance learning due to COVID-19 did not leave teachers and students time to gradually adjust to the new learning situation [36]. As intrinsic motivation is considered particularly relevant in the context of distance education [28, 37], applying the SDT framework to the novel situation of pandemic-induced distance learning may lead to important insights that allow for informed recommendations for teachers and educational institutions about how to proceed in the context of continued distance teaching and learning.

In summary, the COVID-19 situation is a completely new environment, and basic need satisfaction during learning under pandemic-induced conditions has not been explored before. Considering that closures of educational institutions have affected billions of students worldwide and have been strongly debated in some countries, it seems particularly relevant to gain insights into which factors consistently influence conducive or maladaptive learning behavior in these circumstances in a wide range of countries and contextual settings.

Therefore, the overall goal of this study is to investigate the well-established relationship between the three basic needs for autonomy, competence, and social relatedness with intrinsic motivation in the new and specific situation of pandemic-induced distance learning. Firstly, we examine the relationship between each of the basic needs with intrinsic motivation. We

expect that perceived satisfaction of the basic needs for autonomy (H1a), competence (H1b) and social relatedness (H1c) would be positively related to intrinsic motivation. In our second research question, we furthermore extend SDT's predictions regarding two important aspects of learning behavior–procrastination (as a consequence of low or absent intrinsic motivation) and persistence (as the implementation of the volitional part of motivation) and hypothesize that each basic need will be positively related to persistence and negatively related to procrastination, both directly (procrastination: H2a –c; persistence: H3a –c) and mediated by intrinsic motivation (procrastination: H4a –c; persistence: H5a –c). We also proposed that perceived autonomy, competence, and social relatedness would have a direct negative relation with procrastination (H6a –c) and a direct positive relation with persistence (H7a –c). Finally, we investigate SDT's claim of universality, and assume that the aforementioned relationships will emerge across countries we therefore expect a similar pattern of results in all observed countries (H8a –c). As previous studies have indicated that gender [4, 17, 38] and age [39, 40]. May influence intrinsic motivation, persistence, and procrastination, we included participants' gender and age as control variables.

## Methods

### Study design

Due to the circumstances, we opted for a cross-sectional study design across multiple countries, conducted as an online survey. We decided for an online-design due to the pandemic-related restrictions on physical contact with potential survey participants as well as due to the potential to reach a larger audience. As we were interested in the current situation in schools than in long-term development, and we were particularly interested in a large-scale section of the population in multiple countries, we decided on a cross-sectional design. In addition, a multi-country design is particularly interesting in a pandemic setting: During this global health crisis, educational institutions in all countries face the same challenge (to provide distance learning in a way that allows students to succeed) but do so within different frameworks depending on the specific measures each country has implemented. This provides a unique basis for comparing the effects of need fulfillment on students' learning behavior cross-nationally, thus testing the universality of SDT.

### Sample and procedure

The study was carried out across 17 countries, with central coordination taking place in Austria. It was approved and supported by the Austrian Federal Ministry of Education, Science and Research and conducted online. International cooperation partners were recruited from previously established research networks (e.g., European Family Support Network [COST Action 18123]; Transnational Collaboration on Bullying, Migration and Integration at School Level [COST Action 18115]; International Panel on Social), resulting in data collection in 16 countries (Albania, China, Croatia, Estonia, Finland, Germany, Iceland, Japan, Kosovo, Lithuania, Poland, Malta, North Macedonia, Romania, Sweden, USA) in addition to Austria. Data collection was carried out between April and August 2020. During this period, all participating countries were in some degree of pandemic-induced lockdown, which resulted in universities temporarily switching to distance learning. The online questionnaires were distributed among university students via online surveys by the research groups in each respective country. No restrictions were placed on participation other than being enrolled at a university in the sampling country. Participants were informed about the goals of the study, expected time it would take to fill out the questionnaire, voluntariness of participation and anonymity of the acquired

data. All research partners ensured that all ethical and legal requirements related to data collection in their country context were met.

Only data from students who gave their written consent to participate, had reached the age of majority (18 or older) and filled out all questions regarding the study's main variables were included in the analyses (for details on data cleaning rules and exclusion criteria, see [41]). Additional information on data collection in the various countries is provided in S1 Table in S1 File.

The overall sample of $N$ = 15,462 students was predominantly female (71.7%, 27.4% male and 0.7% diverse) and ranged from 18 to 71 years, with the average participant age being 24.41 years ($SD$ = 6.93, $Mdn$ = 22.00). Sample descriptives per country are presented in Table 1.

**Measures.** The variables analyzed here were part of a more extensive questionnaire; the complete questionnaire, as well as the analysis code and the data set, can be found at OSF [42] In order to take the unique situation into account, existing scales were adapted to the current pandemic context (e.g., adding "In the current home-learning situation . . ."), and supplemented with a small number of newly developed items. Subsequently, the survey was revised based on expert judgements from our research group and piloted with cognitive interview testing. The items were sent to the research partners in English and translated separately by each respective research team either using the translation-back-translation method or by at least two native-speaking experts. Subsequently, any differences were discussed, and a consolidated version was established.

To assure the reliability of the scales, we analyzed them using alpha coefficients separately for each country (see S2–S18 Tables in S1 File). All items were answered on a rating scale from 1 (= strongly agree) to 5 (= strongly disagree) and students were instructed to answer with regard to the current situation (distance learning during the COVID-19 lockdown). Analyses were conducted with recoded items so that higher values reflected higher agreement with the statements.

**Table 1. Sociodemographic characteristics of participants per country.**

| Country | N | Gender | | | Age (years) | | | |
|---|---|---|---|---|---|---|---|---|
| | | *female* | *male* | *diverse* | *M* | *SD* | *Mdn* | *Range* |
| Albania | 438 | 354 (80.8%) | 84 (19.2%) | 0 (0%) | 21.53 | 4.326 | 20.0 | 18–50 |
| Austria | 6,071 | 4,167 (68.9%) | 1,858 (30.7%) | 22 (0.4%) | 25.02 | 6.902 | 23.0 | 18–71 |
| China | 404 | 323 (80.0%) | 81 (20.0%) | 0 (0%) | 22.88 | 4.436 | 22.0 | 18–46 |
| Croatia | 330 | 232 (70.3%) | 98 (29.7%) | 0 (0%) | 25.75 | 6.996 | 23.0 | 19–53 |
| Estonia | 104 | 54 (51.9%) | 50 (48.1%) | 0 (0%) | 26.87 | 5.438 | 25.5 | 18–54 |
| Finland | 1,653 | 1,297 (78.5%) | 324 (19.6%) | 32 (1.9%) | 28.49 | 8.934 | 25.0 | 19–69 |
| Germany | 692 | 455 (65.8%) | 228 (32.9%) | 8 (1.2%) | 23.54 | 4.481 | 23.0 | 18–55 |
| Iceland | 348 | 287 (82.5%) | 60 (17.2%) | 1 (0.3%) | 32.06 | 9.645 | 29.0 | 19–71 |
| Japan | 564 | 376 (66.7%) | 183 (32.4%) | 5 (0.9%) | 19.69 | 2.540 | 19.0 | 18–54 |
| Kosovo | 951 | 683 (71.8%) | 265 (27.9%) | 3 (0.3%) | 20.38 | 2.211 | 20.0 | 18–38 |
| Lithuania | 271 | 246 (91.1%) | 23 (8.9%) | 0 (0%) | 25.55 | 7.708 | 22.0 | 18–56 |
| Malta | 201 | 137 (68.8%) | 62 (31.2%) | 0 (0%) | 23.63 | 7.710 | 21.0 | 18–63 |
| North Macedonia | 234 | 197 (84.2%) | 37 (15.8%) | 0 (0%) | 21.79 | 3.731 | 21.0 | 18–57 |
| Poland | 619 | 525 (84.4%) | 90 (14.5%) | 4 (0.6%) | 22.61 | 4.162 | 22.0 | 18–50 |
| Romania | 325 | 245 (75.4%) | 80 (24.6%) | 0 (0%) | 20.86 | 2.698 | 20.0 | 18–41 |
| Sweden | 1,740 | 1,226 (70.55%) | 492 (28.3%) | 22 (1.3%) | - | - | - | - |
| USA | 517 | 276 (54.1%) | 227 (44.5%) | 7 (1.4%) | 20.23 | 2.175 | 20.0 | 18–26 |

In Sweden, age was collected as a categorial variable and is therefore not presented in this table.

*Perceived autonomy* was measured with two newly constructed items ("Currently, I can define my own areas of focus in my studies" and "Currently, I can perform tasks in the way that best suits me"; average α = .78, ranging from .62 to .86).

*Perceived competence* was measured with three items, which were constructed based on the Work-related Basic Need Satisfaction Scale (W-BNS; [25]) and transferred to the learning context ("Currently, I am dealing well with the demands of my studies", "Currently, I have no doubts about whether I am capable of doing well in my studies" and "Currently, I am managing to make progress in studying for university"; average α = .83, ranging from .74 to .91).

*Perceived social relatedness* was assessed with three items, based on the W-BNS [43], ("Currently, I feel connected with my fellow students", "Currently, I feel supported by my fellow students") and the German Basic Psychological Need Satisfaction and Frustration Scale [44]; "Currently, I feel connected with the people who are important to me (family, friends)"; average α = .73, ranging from .64 to .88).

*Intrinsic motivation* was measured with three items which were slightly adapted from the Scales for the Measurement of Motivational Regulation for Learning in University Students (SMR-LS; [45]; "Currently, doing work for university is really fun", "Currently, I am really enjoying studying and doing work for university" and "Currently, I find studying for university really exciting"; average α = .91, ranging from .83 to .94).

*Procrastination* was measured with three items adapted from the Procrastination Questionnaire for Students (Prokrastinationsfragebogen für Studierende; PFS; [46]): "In the current home-learning situation, I postpone tasks until the last minute", "In the current home-learning situation, I often do not manage to start a task when I set out to do so", and "In the current home-learning situation, I only start working on a task when I really need to"; average α = .88, ranging from .74 to .91).

*Persistence* was measured with three items adapted from the EPOCH measure [47]: "In the current home-learning situation, I finish whatever task I begin", "In the current home-learning situation, I keep at my tasks until I am done with them" and "In the current home-learning situation, once I make a plan to study, I stick to it"; average α = .81, ranging from .74 to .88).

**Data analysis.**   Data analyses were conducted using IBM SPSS version 26.0 and Mplus version 8.4. First, we tested for measurement invariance between countries prior to any substantial analyses. We conducted a multigroup confirmatory factor analysis (CFAs) for all scales individually to test for configural, metric, and scalar invariance [48, 49] (see S19 Table in S1 File). We used maximum likelihood parameter estimates with robust standard errors (MLR) to deal with the non-normality of the data. CFI and RMSEA were used as indicators for absolute goodness of model fit. In line with Hu and Bentler [50], the following cutoff scores were considered to reflect excellent and adequate fit to the data, respectively: (a) CFI > 0.95 and CFI > 0.90; (b) RMSEA < .06 and RMSEA < .08. Relative model fit was assessed by comparing BICs of the nested models, with smaller BIC values indicating a better trade-off between model fit and model complexity [51]. Configural invariance indicates a factor structure that is universally applicable to all subgroups in the analysis, metric invariance implies that participants across all groups attribute the same meaning to the latent constructs measured, and scalar invariance indicates that participants across groups attribute the same meaning to the levels of the individual items [51]. Consequently, the extent to which the results can be interpreted depends on the level of measurement invariance that can be established.

For the main analyses, three latent multiple group mediation models were computed, each including one of the basic psychological needs as a predictor, intrinsic motivation as the mediator and procrastination and persistence as the outcomes. These three models served to test the hypothesis that perceived autonomy, competence and social relatedness are related to levels of procrastination and persistence, both directly and mediated through intrinsic

motivation. We used bootstrapping in order to provide analyses robust to non-normal distribution variations, specifying 5,000 bootstrap iterations [52]. Results were estimated using the maximum likelihood (ML) method. Bias-corrected bootstrap confidence intervals are reported.

Finally, in an exploratory step, we investigated the international applicability of the direct and mediated effects. To this end, an additional set of latent mediation models was computed where the path estimates were fixed in order to create an average model across all countries. This was prompted by the consistent patterns of results across countries we observed in the multigroup analyses. Model fit indices of these average models were compared to those of the multigroup models in order to establish the similarity of path coefficients between countries.

## Results

### Statistical prerequisites

Table 2 provides overall descriptive statistics and correlations for all variables (see S2–S18 Tables in S1 File for descriptive statistics for the individual countries).

Metric measurement variance, but not scalar measurement invariance could be established for a simple model including the three individual items and no inter-correlations between perceived competence, perceived social relatedness, intrinsic motivation, and procrastination. For these four variables, the metric invariance model had a good absolute fit, whereas the scalar model did not, due to too high RMSEA; moreover, the relative fit was best for the metric model compared to both the configural and scalar model (see S18 Table in S1 File). Metric, but not scalar invariance could also be established for persistence after modelling residual correlations between items 1 and 2 and items 2 and 3 of the scale. This was necessary due to the similar wording of the items (see "Measures" section for item wordings). Consequently, the same residual correlations were incorporated into all mediation models.

Finally, as the perceived autonomy scale consisted of only two items, it had to be fitted in a model with a correlating factor in order to compute measurement invariance. Both perceived competence and perceived social relatedness were correlated with perceived autonomy ($r$ = .59** and $r$ = .31**, respectively; see Table 2). Therefore, we fit two models combining perceived autonomy with each of these factors; in both cases, metric measurement invariance was established (see S19 Table in S1 File).

**Table 2. Descriptive statistics, correlations, and scale reliabilities (overall sample).**

|  | N | M | SD | 1 | 2 | 3 | 4 | 5 | 6 | 7 | 8 | α |
|---|---|---|---|---|---|---|---|---|---|---|---|---|
| 1 Gender | 15,427 | 1.29 | 0.47 | 1.00 | | | | | | | | |
| 2 Age | 13,722 | 24.41 | 6.93 | .04** | 1.00 | | | | | | | |
| 3 Autonomy | 15,388 | 3.30 | 1.09 | 0.00 | 0.00 | 1.00 | | | | | | .781 |
| 4 Competence | 15,398 | 3.38 | 1.01 | -.02* | .07** | .59** | 1.00 | | | | | .834 |
| 5 Social relatedness | 15,404 | 3.31 | 0.97 | -.10** | -.06** | .31** | .37** | 1.00 | | | | .730 |
| 6 Intrinsic motivation | 15,442 | 3.01 | 1.12 | -.02** | .07** | .56** | .58** | .33** | 1.00 | | | .905 |
| 7 Procrastination | 15,105 | 3.04 | 1.16 | .04** | -.05** | -.25** | -.37** | -.13** | -.31** | 1.00 | | .875 |
| 8 Persistence | 15,037 | 3.33 | 0.94 | .02* | -.03** | .34** | .39** | .22** | .35** | -.46** | 1.00 | .807 |

The sample size for age is smaller because in Sweden, age was collected as a categorial variable. Age was therefore recoded into the same categories in all countries so it could be included as a control variable. For the purpose of presenting descriptive statistics of the sample, however, the non-categorial variable is better suited.

*$p$ < .05.

**$p$ < .01.

In summary, these results suggest that the meaning of all constructs we aimed to measure was understood similarly by participants across different countries. Consequently, we were able to fit the same mediation model in all countries and compare the resulting path coefficients.

Both gender and age were statistically significantly correlated with perceived competence, perceived social relatedness, intrinsic motivation, procrastination, and persistence (see S20–S22 Tables in S1 File).

## Mediation analyses

**Autonomy hypothesis.** We hypothesized that higher perceived autonomy would relate to less procrastination and more persistence, both directly and indirectly (mediated through intrinsic learning motivation). Indeed, perceived autonomy was related negatively to procrastination (H6a) in most countries. Confidence intervals did not include zero in 10 out of 17 countries, all effect estimates were negative and standardized effect estimates ranged from $b_{stand}$ = -.02 to -.46 (see Fig 1). Furthermore, perceived autonomy was directly positively related to persistence in most countries. Specifically, for the direct effect of perceived autonomy on persistence (H7a), all but one country (USA, $b_{stand}$ = -.02; $p$ = .621; CI [-.13, .08]) exhibited distinctly positive effect estimates ranging from $b_{stand}$ = .18 to .72 and confidence intervals that did not include zero.

In terms of indirect effects of perceived autonomy on procrastination mediated by intrinsic motivation (H7a), confidence intervals did not include zero in 8 out of 17 countries and effect estimates were mostly negative, ranging from $b_{stand}$ = -.33 to .03. Indirect effects of perceived autonomy on persistence (mediated by intrinsic motivation; H5a) were distinctly positive and confidence intervals did not include zero in 12 out of 17 countries. The indirect effect estimates and confidence intervals for all remaining countries were consistently positive, with the

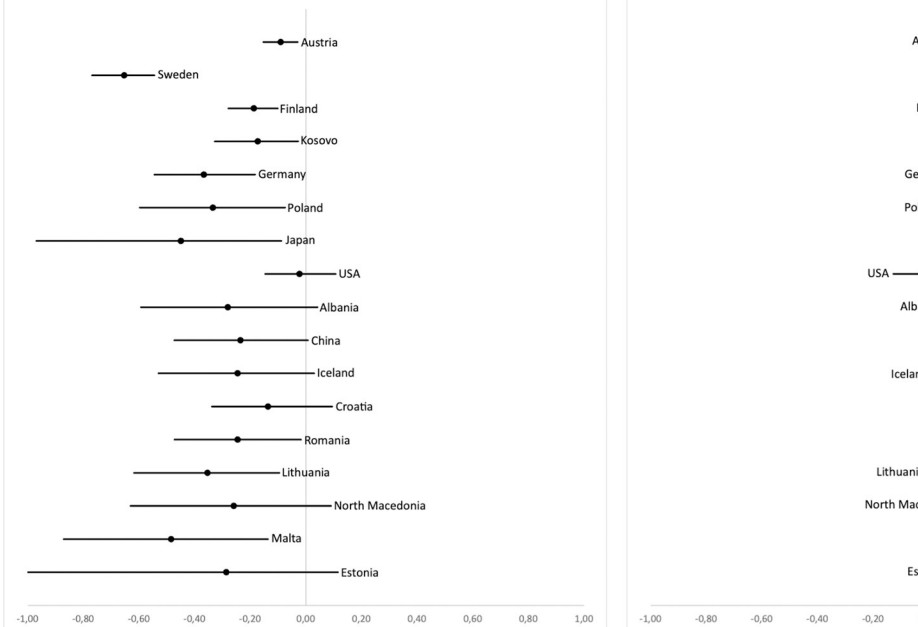

**Fig 1. Unstandardized coefficients and two-sided 5% confidence intervals for the direct effects of perceived autonomy on procrastination (left) and persistence (right).** Countries are ordered by sample size from top (highest) to bottom (lowest).

Indirect effects perceived autonomy - intrinsic motivation - procrastination          Indirect effects perceived autonomy - intrinsic motivation - persistence

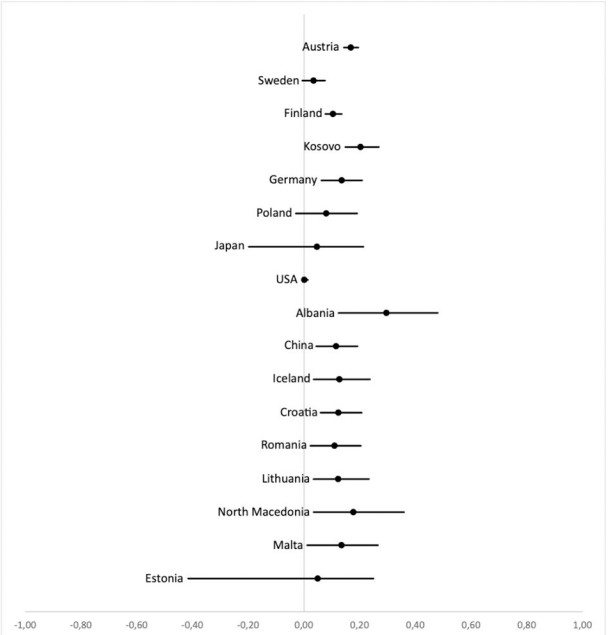

**Fig 2. Unstandardized coefficients and two-sided 5% confidence intervals for the indirect effects of perceived autonomy on procrastination (left) and persistence (right), mediated by intrinsic motivation.** Countries are ordered by sample size from top (highest) to bottom (lowest).

standardized effect estimates ranging from $b_{stand}$ = .13 to .39, indicating a robust, positive mediated effect of autonomy on persistence. Fig 2 displays the unstandardized path coefficients and their two-sided 5% confidence intervals for the indirect effects of perceived autonomy on procrastination via intrinsic motivation (left) and of perceived autonomy on persistence via intrinsic motivation (right).

Unstandardized and standardized path coefficients, standard errors, p-values and bias-corrected bootstrapped confidence intervals for the direct and indirect effects of perceived autonomy on procrastination and persistence for each country are provided in S23–S26 Tables in S1 File, respectively.

**Competence hypothesis.** Secondly, we hypothesized that higher perceived competence would relate to less procrastination and more persistence both directly and indirectly, mediated through intrinsic learning motivation. Direct effects on procrastination (H6b) were negative in most countries and confidence intervals did not include zero in 10 out of 17 countries (see Fig 3).

Standardized effect estimates ranged from $b_{stand}$ = -.02 to -.60, with 10 out of 17 countries exhibiting at least a medium-sized effect. Correspondingly, effect estimates for the direct effects on persistence were positive everywhere except the USA and confidence intervals did not include zero in 14 out of 17 countries (see Fig 3). Standardized effect estimates ranged from $b_{stand}$ = -.05 to .64 with 14 out of 17 countries displaying an at least medium-sized positive effect.

The pattern of results for the indirect effects of perceived competence on procrastination mediated by learning motivation (H4b) is illustrated in Fig 4: Effect estimates were negative with the exception of China and the USA. Confidence intervals did not include zero in 7 out of 17 countries. Standardized effect estimates range between $b_{stand}$ = .06 and -.46. Indirect effects of perceived competence on persistence were positive everywhere except for two countries and

Direct effects perceived competence - procrastination Direct effects perceived competence - persistence

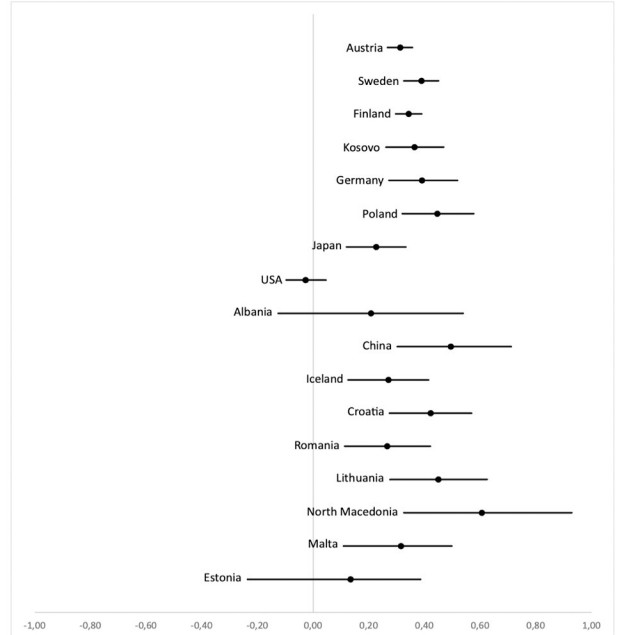

**Fig 3. Unstandardized coefficients and two-sided 5% confidence intervals for the direct effects of perceived competence on procrastination (left) and persistence (right).** Countries are ordered by sample size from top (highest) to bottom (lowest).

Indirect effects perceived competence - intrinsic motivation - procrastination Indirect effects perceived competence - intrinsic motivation - persistence

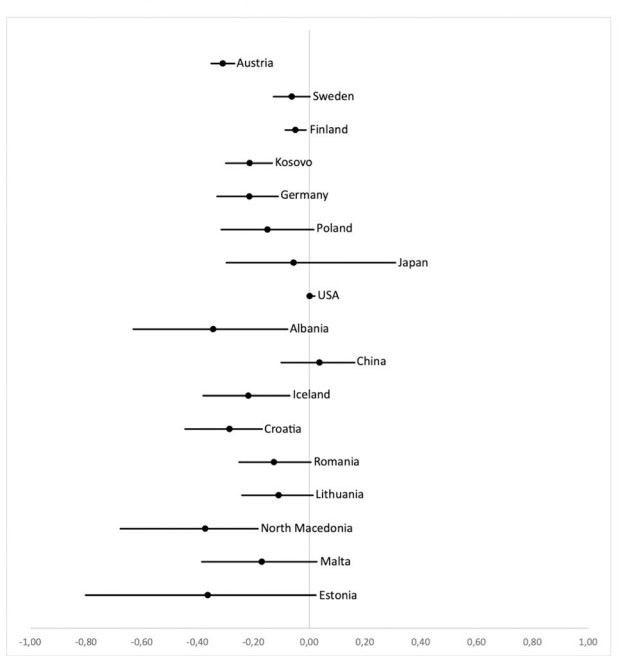 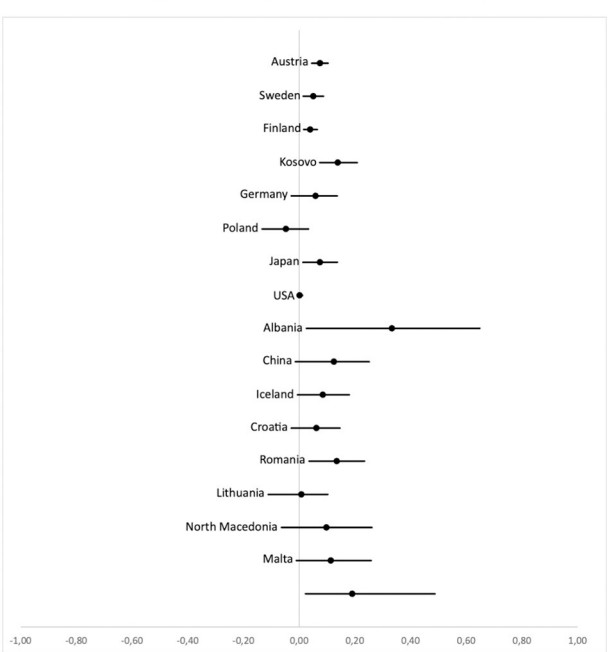

**Fig 4. Unstandardized coefficients and two-sided 5% confidence intervals for the indirect effects of perceived competence on procrastination (left) and persistence (right), mediated by intrinsic motivation.** Countries are ordered by sample size from top (highest) to bottom (lowest).

Direct effects perceived social relatedness - procrastination | Direct effects perceived social relatedness - persistence

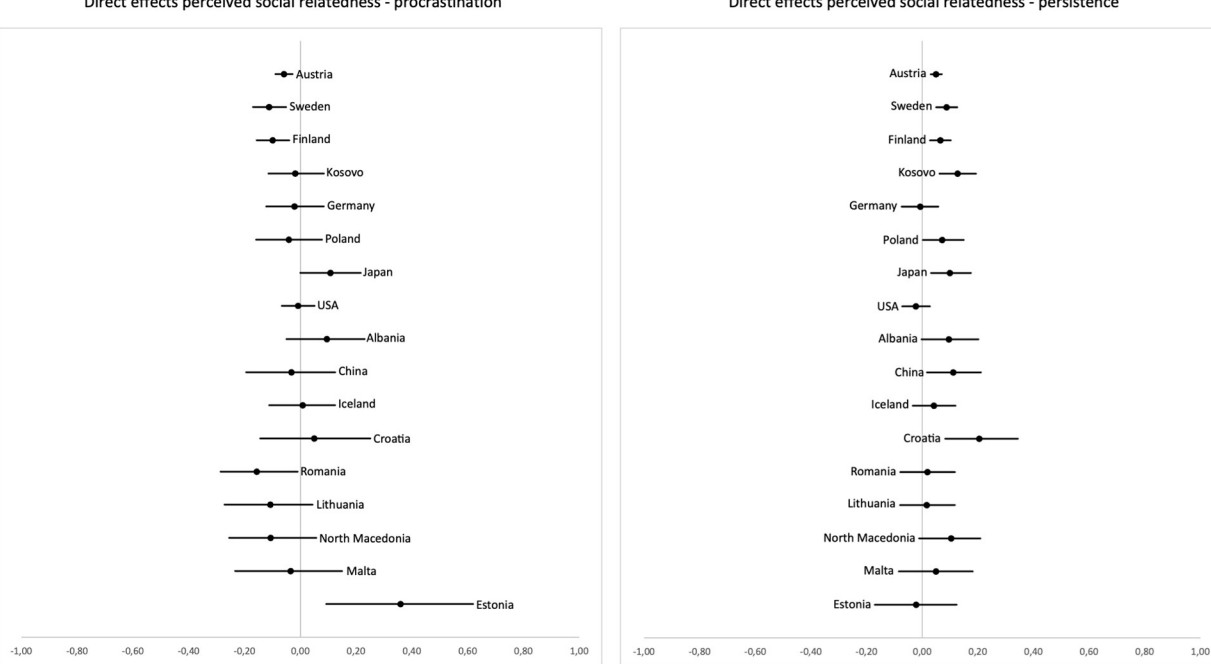

**Fig 5. Unstandardized coefficients and two-sided 5% confidence intervals for the direct effects of perceived social relatedness on procrastination (left) and persistence (right).** Countries are ordered by sample size from top (highest) to bottom (lowest).

confidence intervals did not include zero in 7 out of 17 countries (see Fig 4). Standardized effect estimates varied between $b_{stand}$ = -.07 and .46 (see S23–S26 Tables in S1 File for unstandardized and standardized path coefficients).

**Social relatedness hypothesis.** Finally, we hypothesized that stronger perceived social relatedness would be both directly and indirectly (mediated through intrinsic learning motivation) related to less procrastination and more persistence. The pattern of results was more ambiguous here than for perceived autonomy and perceived competence. Direct effect estimates on procrastination (H6c) were negative in 12 countries; however, the confidence intervals included zero in 12 out of 17 countries (see Fig 5). Standardized effect estimates ranged from $b_{stand}$ = -.01 to $b_{stand}$ = .33. The direct relation between perceived social relatedness and persistence (H7c) yielded 14 negative and three positive effect estimates. Confidence intervals did not include zero in 7 out of 17 countries (see Fig 5), with standardized effect estimates ranging from $b_{stand}$ = -.01 to $b_{stand}$ = .31.

In terms of indirect effects of perceived social relatedness being related to procrastination mediated by intrinsic motivation (H4c), the pattern of results was consistent: All effect estimates except those for the USA were clearly negative, and confidence intervals did not include zero in 15 out of 17 countries (see Fig 6). Standardized effect estimates ranged between $b_{stand}$ = .00 and $b_{stand}$ = -.46. Indirect paths of perceived social relatedness on persistence showed positive effect estimates and standardized effect estimates ranging from $b_{stand}$ = .00 to .44 and confidence intervals not including zero in 16 out of 17 countries (see Fig 6; see S23–S26 Tables in S1 File for unstandardized and standardized path coefficients).

## Meta-analytic approach

Due to the overall similarity of the results across many countries, we decided to compute, in an additional, exploratory step, the same models with path estimates fixed across countries.

Indirect effects perceived social relatedness - intrinsic motivation - procrastination

Indirect effects perceived social relatedness - intrinsic motivation - persistence

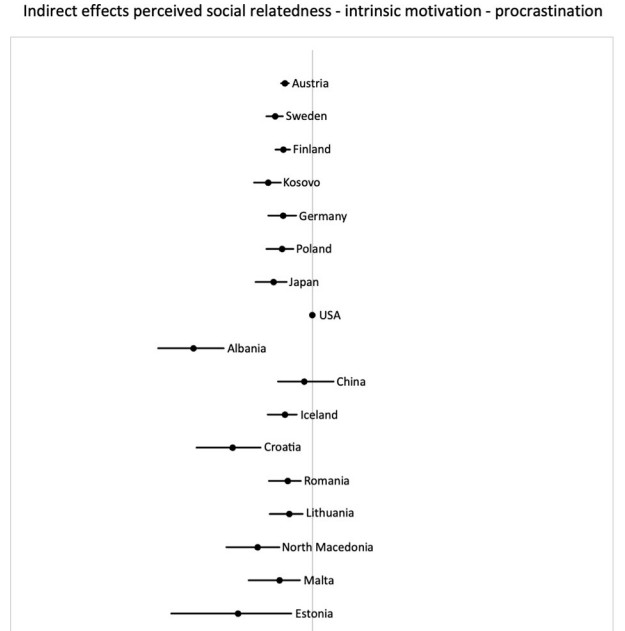
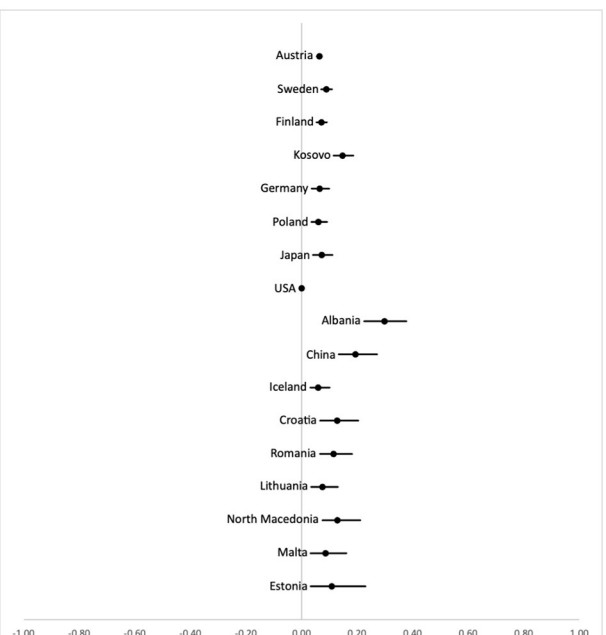

**Fig 6. Unstandardized coefficients and two-sided 5% confidence intervals for the indirect effects of perceived social relatedness on procrastination (left) and persistence (right), mediated by intrinsic motivation.** Countries are ordered by sample size from top (highest) to bottom (lowest).

This resulted in three models with average path estimates across the entire sample. Standardized path coefficients for the direct and indirect effects of the basic psychological needs on procrastination and persistence are presented in S27 and S28 Tables in S1 File, respectively. We compared the model fits of these three average models to those of the multigroup mediation models: If the fit of the average model is better than that of the multigroup model, it indicates that the individual countries are similar enough to be combined into one model. The amount of explained variance per model, outcome variable and country are provided in S29 Table in S1 File for procrastination and S30 Table in S1 File for persistence.

**Perceived autonomy.** Relative model fit was better for the perceived autonomy model with fixed paths (BIC = 432,707.89) compared to the multigroup model (BIC = 432,799.01). Absolute model fit was equally good in the multigroup model (RMSEA = 0.05, CFI = 0.98, TLI = 0.97) and in the fixed path model (RMSEA = 0.05, CFI = 0.97, TLI = 0.97). Consequently, the general model in Fig 7 describes the data from all 17 countries equally well. The average amount of explained variance, however, is slightly higher in the multigroup model, with 19.9% of the variance in procrastination and 33.7% of the variance in persistence explained, as compared to 18.3% and 27.6% in the fixed path model. The amount of variance explained increased substantially in some countries when fixing the paths: in the multigroup model, explained variance ranges from 2.2% to 44.4% for procrastination and from 0.9% to 69.9% for persistence, compared to 13.0% - 27.7% and 18.2% to 63.2% in the fixed path model. Notably, the amount of variance explained did not change much in the three countries with the largest samples, Austria, Sweden, and Finland; countries with much smaller samples and larger confidence intervals were more affected.

Overall, perceived autonomy had significant direct and indirect effects on both procrastination and persistence; higher perceived autonomy was related to less procrastination directly

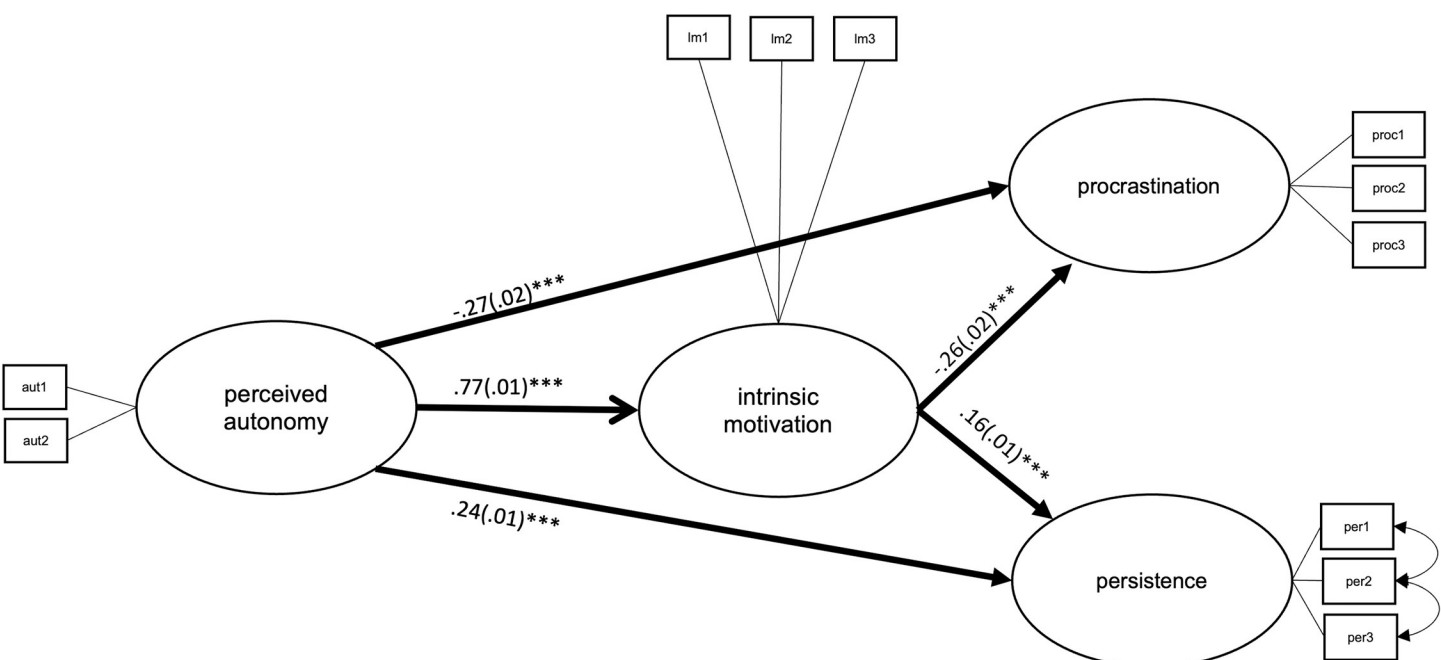

**Fig 7. Latent structural path model of the direct effects between perceived autonomy, intrinsic motivation, procrastination, and persistence, with average coefficients across all countries.** ***$p = < .001$.

($b_{\text{unstand}}$ = -.27, $SE$ = .02, $p = < .001$) and mediated by learning motivation ($b_{\text{unstand}}$ = -.20, $SE$ = .01, $p = < .001$) and to more persistence directly ($b_{\text{unstand}}$ = .24, $SE$ = .01, $p = < .001$) and mediated by learning motivation ($b_{\text{unstand}}$ = .12, $SE$ = .01, $p = < .001$). Direct effects for the autonomy model are shown in Fig 7; for the indirect effects see Table 3.

Effects of age and gender varied across countries (see S20 Table in S1 File).

**Perceived competence.** For the perceived competence model, relative fit decreased when fixing the path coefficient estimates (BIC = 465,830.44 to BIC = 466,020.70). The absolute fit indices were also better for the multigroup model (RMSEA = 0.05, CFI = 0.97, TLI = 0.96) than for the fixed path model (RMSEA = 0.06, CFI = 0.96, TLI = 0.96). Hence, multigroup modelling describes the data across all countries somewhat better than a fixed path model as depicted in Fig 8. Correspondingly, the fixed path model explained less variance on average than did the multigroup model, with 23.2% instead of 24.3% of the variance in procrastination and 32.9% instead of 37.3% of the variance in persistence explained. Explained variance ranged from 1.0% to 51.9% for procrastination in the multigroup model, as compared to 13.9% - 34.4% in the fixed path model. The amount of variance in persistence explained ranged from 1.0% to 58.1% in the multigroup model and from 23.5% to 55.9% in the fixed path model (see S29 and S30 Tables in S1 File).

**Table 3. Indirect effects of the three basic psychological needs on procrastination and persistence in the fixed path model.**

| | Procrastination | | | | Persistence | | | |
|---|---|---|---|---|---|---|---|---|
| | Est. | SE | p | CI | Est. | SE | p | CI |
| Autonomy | -0.20 | 0.01 | < .001 | (-0.23\|-0.18) | 0.12 | 0.01 | < .001 | (0.11\|0.13) |
| Competence | -0.11 | 0.01 | < .001 | (-0.13\|-0.09) | 0.07 | 0.01 | < .001 | (-0.13\|-0.11) |
| Social relatedness | -0.12 | 0.01 | < .001 | (0.06\|0.09) | 0.08 | 0.00 | < .001 | (0.07\|0.09) |

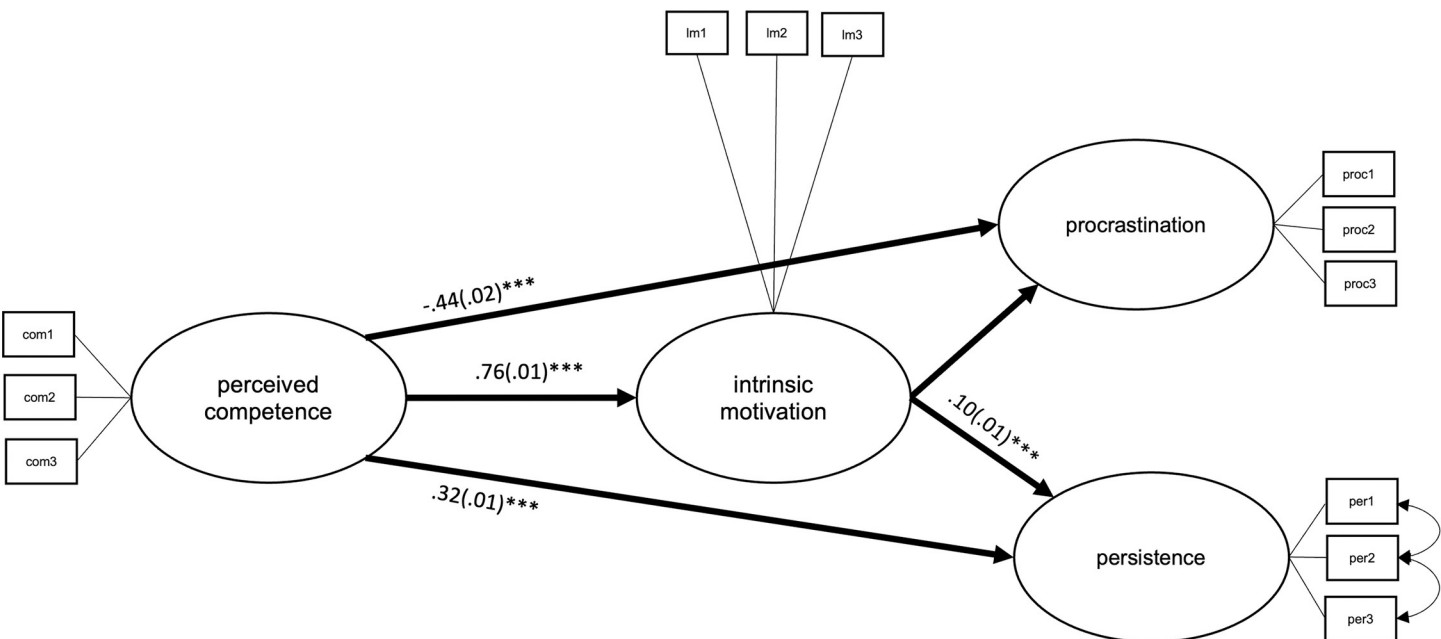

**Fig 8. Latent structural path model of the direct effects between perceived competence, intrinsic motivation, procrastination, and persistence, with average coefficients across all countries.** ***$p = < .001$.

Overall, higher perceived competence was related to less procrastination ($b_{unstand}$ = -.44, $SE$ = .02, $p = < .001$) and to higher persistence ($b_{unstand}$ = .32, $SE$ = .01, $p = < .001$). These effects were partly mediated by intrinsic learning motivation ($b_{unstand}$ = -.11, $SE$ = .01, $p = < .001$, and $b_{unstand}$ = .07, $SE$ = .01, $p = < .001$, respectively; see Table 3). Effects of gender and age varied between countries, see S21 Table in S1 File.

**Perceived social relatedness.** Finally, the perceived social relatedness model with fixed paths had a relatively better model fit (BIC = 479,428.46) than the multigroup model (BIC = 479,604.61). Likewise, the absolute model fit was similar in the model with path coefficients fixed across countries (RMSEA = 0.05, CFI = 0.97, TLI = 0.96) and the multigroup model (RMSEA = 0.05, CFI = 0.97, TLI = 0.97). The multigroup model explained 17.6% of the variance in procrastination and 26.3% of the variance in persistence, as compared to 15.2% and 21.6%, respectively in the fixed path model. Explained variance for procrastination ranged between 0.5% and 48.1% in the multigroup model, and from 9.0% to 23.0% in the fixed path model. Similarly, the multigroup model explained between 1.0% and 56.5% of the variance in persistence across countries, while the fixed path model explained between 15.6% and 48.3% (see S29 and S30 Tables in S1 File).

Hence, the fixed path model depicted in Fig 9 is well-suited for describing data across all 17 countries. Higher perceived social relatedness is related to less procrastination both directly ($b_{unstand}$ = -.06, $SE$ = .01, $p = < .001$) and indirectly through learning motivation ($b_{unstand}$ = -.12, $SE$ = .01, $p = < .001$). Likewise, it is related to higher persistence both directly ($b_{unstand}$ = .07, $SE$ = .01, $p = < .001$) and indirectly through learning motivation ($b_{unstand}$ = .08, $SE$ = .00, $p = < .001$; see Table 3). Effects of gender and age are shown in S22 Table in S1 File.

## Discussion

The aim of this study was to extend current research on the association between the basic psychological needs for autonomy, competence, and social relatedness with intrinsic motivation and two important aspects of learning behavior—procrastination and persistence—in the new

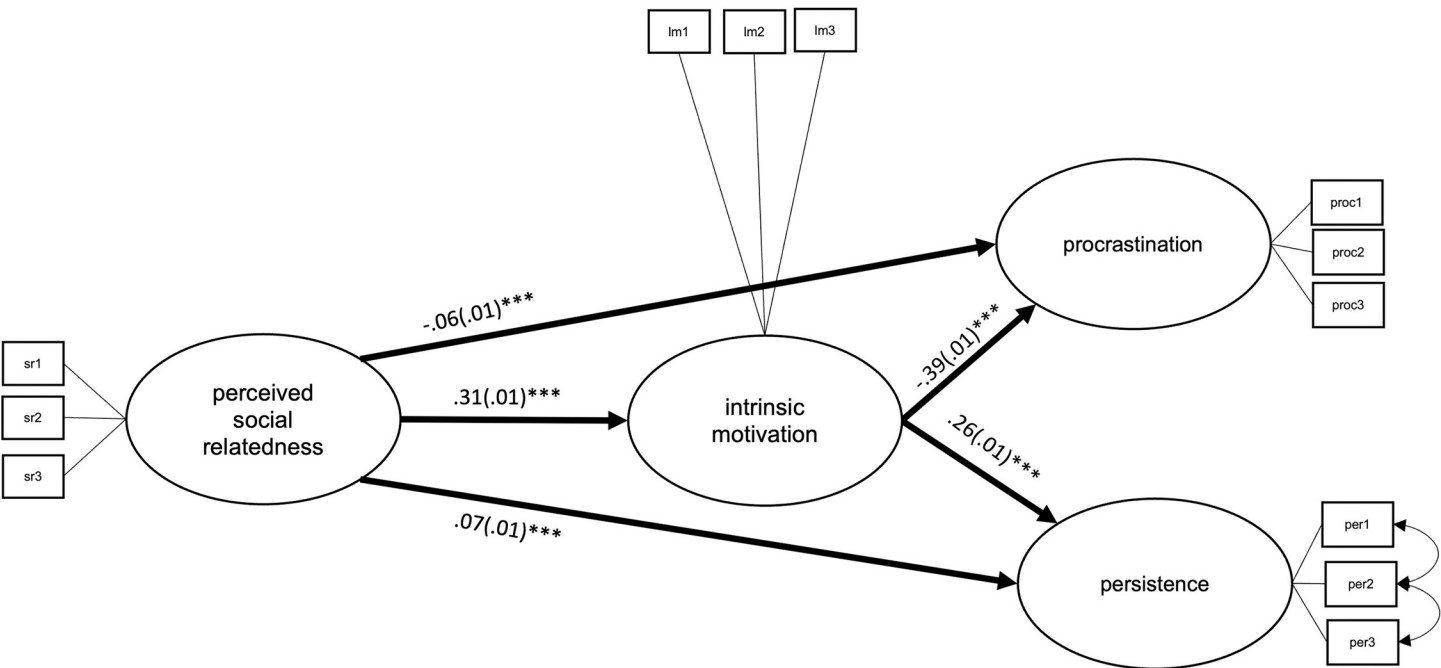

**Fig 9. Latent structural path model of the direct effects between perceived social relatedness, intrinsic motivation, procrastination, and persistence, with average coefficients across all countries.** ***$p = < .001$.

and unique situation of pandemic-induced distance learning. We also investigated SDT's [7] postulate that the relation between basic psychological need satisfaction and active (persistence) as well as passive (procrastination) learning behavior is mediated by intrinsic motivation. To test the theory's underlying claim of universality, we collected data from $N = 15,462$ students across 17 countries in Europe, Asia, and North America.

Confirming our hypothesis, we found that the three basic psychological needs were consistently and positively related to intrinsic motivation in all countries except for the USA (H1a - c). This consistent result is in line with self-determination theory [7] and other previous studies (e.g., 9), which have found that satisfaction of the three basic needs for autonomy, competence and social relatedness is related to higher intrinsic motivation. Notably, the association with intrinsic motivation was stronger for perceived autonomy and perceived competence than for perceived social relatedness. This also has been found in previous studies [4, 9, 28]. Pandemic-induced distance learning, where physical and subsequential social contact in all areas of life was severely constricted, might further exacerbate this discrepancy, as instructors may have not been able to establish adequate communication structures due to the rapid switch to distance learning [36, 53]. As hypothesized, intrinsic motivation was in general negatively related to procrastination (H2a - c) and positively related to persistence (H3a - c), indicating that students who are intrinsically motivated are less prone to procrastination and more persistent when studying. This again underlines the importance of intrinsic motivation for adaptive learning behavior, even and particularly in a distance learning setting, where students are more prone to disengage from classes [34].

### The mediating effect of intrinsic motivation on procrastination and persistence

Direct effects of the basic needs on the outcomes were consistently more ambiguous (with smaller effect estimates and larger confidence intervals, including zero in more countries) than

indirect effects mediated by intrinsic motivation. This difference was particularly pronounced for perceived social relatedness, where a clear negative direct effect on procrastination (H6c) could be observed only in the three countries with the largest sample size (Austria, Sweden, Finland) and Romania, whereas the confidence interval in most countries included zero. Moreover, in Estonia there was even a clear positive effect. The unexpected effect in the Estonian sample may be attributed to the fact that this country collected data only from international exchange students. Since the lockdown in Estonia was declared only a few weeks after the start of the semester, international exchange students had only a very short period of time to establish contacts with fellow students on site. Accordingly, there was probably little integration into university structures and social contacts were maintained more on a personal level with contacts from the home country. Thus, such students' fulfillment of this basic need might have required more time and effort, leading to higher procrastination and less persistence in learning.

A diametrically opposite pattern was observed for persistence (H7c), where some direct effects of social relatedness were unexpectedly negative or close to zero. We therefore conclude that evidence for a direct negative relationship between social relatedness and procrastination and a direct positive relationship between social relatedness and persistence is lacking. This could be due to the specificity of the COVID-19 situation and resulting lockdowns, in which maintaining social contact took students' focus off learning. In line with SDT, however, indirect effects of perceived social relatedness on procrastination (H4c) and persistence (H5c) mediated via intrinsic motivation were much more visible and in the expected directions. We conclude that, while the direct relation between perceived social relatedness and procrastination is ambiguous, there is strong evidence that the relationship between social relatedness and the measured learning behaviors is mediated by intrinsic motivation. Our results strongly underscore SDT's assumption that close social relations promote intrinsic motivation, which in turn has a positive effect on learning behavior (e.g., [6, 14]). The effects for perceived competence exhibited a somewhat clearer and hypothesis-conforming pattern. All direct effects of perceived competence on procrastination (H6b) were in the expected negative direction, albeit with confidence intervals spanning zero in 7 out of 17 countries. Direct effects of perceived competence on persistence (H7b) were consistently positive with the exception of the USA, where we observed a very small and non-significant negative effect. Indirect effects of perceived competence on procrastination (H4b) and persistence (H5b) as mediated by intrinsic motivation were mostly consistent with our expectations as well. Considering this overall pattern of results, we conclude that there is strong evidence that perceived competence is negatively associated with procrastination and positively associated with persistence. Furthermore, our results also support SDT's postulate that the relationship between perceived competence and the measured learning behaviors is mediated by intrinsic motivation.

It is notable that the estimated direct effects of perceived competence on procrastination and persistence were higher than the indirect effects in most countries we investigated. Although SDT proposes that perceived competence leads to higher intrinsic motivation, Deci and Ryan [8] also argue that it affects all types of motivation and regulation, including less autonomous forms such as introjected and identified motivation, indicating that if the need for competence is not satisfied, all types of motivation are negatively affected. This may result in a general amotivation and lack of action. In our study, we only investigated intrinsic motivation as a mediator. For future research, it might be advantageous to further differentiate between different types of externally and internally controlled behavior. Furthermore, perceived competence increases when tasks are experienced as optimally challenging [7, 54]. However, in order for instructors to provide the optimal level of difficulty and support needed, frequent communication with students is essential. Considering that data collection for the

present study took place at a time of great uncertainty, when many countries had only transitioned to distance learning a few weeks prior, it is reasonable to assume that both structural support as well as communication and feedback mechanisms had not yet matured to a degree that would favor individualized and competency-based work.

However, our findings corroborate those from earlier studies insofar as they underline the associations between perceived competence and positive learning behavior (e.g., [19]), that is, lower procrastination [18] and higher persistence (e.g., [21]), even in an exceptional situation like pandemic-induced distance learning.

Turning to perceived autonomy, although the confidence intervals for the direct effects of perceived autonomy on procrastination (H6a) did span zero in most countries with smaller sample sizes, all effect estimates indicated a negative relation with procrastination. We expected these relationships from previous studies [18, 23]; however, the effect might have been even more pronounced in the relatively autonomous learning situation of distance learning, where students usually have increased autonomy in deciding when, where, and how to learn. While this bears the risk of procrastination, it also comes with the opportunity to consciously delay less pressing tasks in favor of other, more important or urgent tasks (also called *strategic delay*) [5], resulting in lower procrastination. In future studies, it might be beneficial to differentiate between passive forms of procrastination and active strategic delay in order to obtain more detailed information on the mechanisms behind this relationship. Direct effects of autonomy on persistence (H7a) were consistently positive. Students who are free to choose their preferred time and place to study may engage more with their studies and therefore be more persistent.

Indirect effects of perceived autonomy on procrastination mediated by intrinsic motivation (H4a) were negative in all but two countries (China and the USA), which is generally consistent with our hypothesis and in line with previous research (e.g., [23]). Additionally, we found a positive indirect effect of autonomy on persistence (H5a), indicating that autonomy and intrinsic motivation play a crucial role in students' persistence in a distance learning setting. Based on our results, we conclude that perceived autonomy is negatively related to procrastination and positively related to persistence, and that this relationship is mediated by intrinsic motivation. It is worth noting that, unlike with perceived competence, the direct and indirect effects of perceived autonomy on the outcomes procrastination and persistence were similarly strong, suggesting that perceived autonomy is important not only as a driver of intrinsic motivation but also at a more direct level. It is important to make the best possible use of the opportunity for greater autonomy that distance learning offers. However, autonomy is not to be equated with a lack of structure; instead, learners should be given the opportunity to make their own decisions within certain framework conditions.

## The applicability of self-determination theory across countries

Overall, the results of our mediation analysis for the separate countries support the claim posited by SDT that basic need satisfaction is essential for intrinsic motivation and learning across different countries and settings. In an exploratory analysis, we tested a fixed path model including all countries at once, in order to test whether a simplified general model would yield a similar amount of explained variance. For perceived autonomy and social relatedness, the model fit increased, whereas for perceived competence it decreased slightly compared to the multigroup model. However, all fixed path models exhibited adequate model fit. Considering that the circumstances in which distance learning took place in different countries varied to some degree (see also Limitations), these findings are a strong indicator for the universality of SDT.

### Study strengths and limitations

Although the current study has several strengths, including a large sample size and data from multiple countries, three limitations must be considered. First, it must be noted that sample sizes varied widely across the 17 countries in our study, with one country above 6,000 (Austria), two above 1,000 (Finland and Sweden) and the rest ranging between 104 and 905. Random sampling effects are more problematic in smaller samples; hence, this large variation weakens our ability to conduct cross-country comparisons. At the same time, small sample sizes weaken the interpretability of results within each country; thus, our results for Austria, Finland and Sweden are considerably more robust than for the remaining fourteen countries. Additionally, two participating countries collected specific subsamples: In China, participants were only recruited from one university, a nursing school. In Estonia, only international exchange students were invited to participate. Nevertheless, with the exception of the unexpected positive direct relationship between social relatedness and procrastination, all observed divergent effects were non-significant. Indeed, this adds to the support for SDT's claims to universality regarding the relationship between perceived autonomy, competence, and social relatedness with intrinsic motivation: Results in the included countries were, despite their differing subsamples, in line with the overall trend of results, supporting the idea that SDT applies equally to different groups of learners.

Second, due to the large number of countries in our sample and the overall volatility of the situation, learning circumstances were not identical for all participants. Due to factors such as COVID-19 case counts and national governments' political priorities, lockdown measures varied in their strictness across settings. Some universities were fully closed, some allowed on-site teaching for particular groups (e.g., students in the middle of a laboratory internship), and some switched to distance learning but held exams on site (see S1 Table in S1 File for further information). Therefore, learning conditions were not as comparable as in a strict experimental setting. On the other hand, this strengthens the ecological validity of our study. The fact that the pattern of results was similar across contexts with certain variation in learning conditions further supports the universal applicability of SDT.

Finally, due to the novelty of the COVID-19 situation, some of the measures were newly developed for this study. Due to the need to react swiftly and collect data on the constantly evolving situation, it was not possible to conduct a comprehensive validation study of the instruments. Nevertheless, we were able to confirm the validity of our instruments in several ways, including cognitive interview testing, CFAs, CR, and measurement invariance testing.

## Conclusion and future directions

In general, our results further support previous research on the relation between basic psychological need fulfilment and intrinsic motivation, as proposed in self-determination theory. It also extends past findings by applying this well-established theory to the new and unique situation of pandemic-induced distance learning across 17 different countries. Moreover, it underlines the importance of perceived autonomy and competence for procrastination and persistence in this setting. However, various other directions for further research remain to be pursued. While our findings point to the relevance of social relatedness for intrinsic motivation in addition to perceived competence and autonomy, further research should explore the specific mechanisms necessary to promote social connectedness in distance learning. Furthermore, in our study, we investigated intrinsic motivation, as the most autonomous form of motivation. Future research might address different types of externally and internally regulated motivation in order to further differentiate our results regarding the relations between basic need satisfaction and motivation. Finally, a longitudinal study design could provide deeper

insights into the trajectory of need satisfaction, intrinsic motivation and learning behavior during extended periods of social distancing and could provide insights into potential forms of support implemented by teachers and coping mechanisms developed by students.

## Supporting information

**S1 File.**
(DOCX)

## Author Contributions

**Conceptualization:** Elisabeth R. Pelikan, Selma Korlat, Julia Holzer, Barbara Schober, Christiane Spiel, Marko Lüftenegger.

**Data curation:** Selma Korlat, Martin Mayerhofer, Oriola Hamzallari, Ana Uka, Jiarui Chen, Maritta Välimäki, Zrinka Puharić, Kelechi Evans Anusionwu, Angela Nkem Okocha, Anastassia Zabrodskaja, Katariina Salmela-Aro, Udo Käser, Anja Schultze-Krumbholz, Sebastian Wachs, Finnur Friðriksson, Hermína Gunnþórsdóttir, Yvonne Höller, Ikuko Aoyama, Akihiko Ieshima, Yuichi Toda, Jon Konjufca, Njomza Llullaku, Reda Gedutienė, Glorianne Borg Axisa, Irena Avirovic Bundalevska, Angelka Keskinova, Makedonka Radulovic, Aleksandra Lewandowska-Walter, Justyna Michałek-Kwiecień, Piotr Plichta, Jacek Pyżalski, Natalia Walter, Cristina Cautisanu, Ana Iolanda Voda, Shang Gao, Sirajul Islam, Kai Wistrand, Michelle F. Wright.

**Formal analysis:** Julia Reiter.

**Funding acquisition:** Barbara Schober, Christiane Spiel, Marko Lüftenegger.

**Methodology:** Elisabeth R. Pelikan, Selma Korlat, Julia Reiter, Julia Holzer, Barbara Schober, Christiane Spiel, Marko Lüftenegger.

**Project administration:** Selma Korlat.

**Writing – original draft:** Elisabeth R. Pelikan, Julia Reiter.

**Writing – review & editing:** Selma Korlat, Julia Reiter, Julia Holzer, Martin Mayerhofer, Barbara Schober, Oriola Hamzallari, Ana Uka, Jiarui Chen, Maritta Välimäki, Zrinka Puharić, Kelechi Evans Anusionwu, Angela Nkem Okocha, Anastassia Zabrodskaja, Katariina Salmela-Aro, Udo Käser, Anja Schultze-Krumbholz, Sebastian Wachs, Finnur Friðriksson, Hermína Gunnþórsdóttir, Yvonne Höller, Ikuko Aoyama, Akihiko Ieshima, Yuichi Toda, Jon Konjufca, Njomza Llullaku, Reda Gedutienė, Glorianne Borg Axisa, Irena Avirovic Bundalevska, Angelka Keskinova, Makedonka Radulovic, Aleksandra Lewandowska-Walter, Justyna Michałek-Kwiecień, Piotr Plichta, Jacek Pyżalski, Natalia Walter, Cristina Cautisanu, Ana Iolanda Voda, Shang Gao, Sirajul Islam, Kai Wistrand, Michelle F. Wright, Marko Lüftenegger.

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
