## [Decision Letter · Decision Letter 0]

26 May 2021

PONE-D-21-10408

Distance learning in higher education during COVID-19: The role of basic psychological needs and intrinsic motivation for persistence and procrastination – A multi-country study

PLOS ONE

Dear Dr. Pelikan,

Thank you for submitting your manuscript to PLOS ONE. After careful consideration, we feel that it has merit but does not fully meet PLOS ONE’s publication criteria as it currently stands. Therefore, we invite you to submit a revised version of the manuscript that addresses the points raised during the review process.

The study is very interesting and adjusts to the thematic priorities of the journal's editorial policy, but it requires complementary work.

Two expert reviewers have reviewed the paper and suggest that a number of changes be addressed.

We look forward to receiving your revised manuscript.

Kind regards,

José Gutiérrez-Pérez

Academic Editor

PLOS ONE

Journal Requirements:

3. We noted in your submission details that a portion of your manuscript may have been presented or published elsewhere.

[The data for this manuscript were collected as part of a larger project on learning under COVID-19 conditions led by the Austrian research team. Some variables from 2 of the 15 data sets (Austria and Finland) were analyzed for another paper focused on the relationship of basic need satisfaction with self-regulated learning and student well-being, published by AERA Open under the title "Higher Education in Times of COVID-19: University Students' Basic Need Satisfaction, Self-Regulated Learning and Well-Being."]

**Comments to the Author**

1. Is the manuscript technically sound, and do the data support the conclusions?

Reviewer #1: Yes

Reviewer #2: Partly

2. Has the statistical analysis been performed appropriately and rigorously? 

Reviewer #1: Yes

Reviewer #2: Yes

3. Have the authors made all data underlying the findings in their manuscript fully available?

Reviewer #1: Yes

Reviewer #2: Yes

4. Is the manuscript presented in an intelligible fashion and written in standard English?

Reviewer #1: Yes

Reviewer #2: Yes

5. Review Comments to the Author

Reviewer #1: In one of the initial pages of the material, 17 countries are mentioned, but in reality there are 16.

Clarify the reason why there are no Latin American countries.

Indicate to what the authors attribute the higher number of female vs. male participants.

Point out the reason why Austria, Finland and Switzerland have a higher number of participants.

Not mentioned in the instrument What were the items added to adapt to the pandemic?

Reviewer #2: This is a very interesting, multi-author, multi-country study, which aims to investigate the relationship between basic need satisfaction and procrastination and persistence in the context of emergency distance learning during the COVID-19 pandemic.

This is an interesting paper and the data can be used by other researchers in the field in different countries.

Below, please find some proposals for improvement:

a) The abstract shouldn't include statistical data and such detail on the participants.

b) The "present study" section should be deleted. Important information about the objectives and the methodology of the research can be placed in the introduction and the methods sections respectively.

c) I missed explicit presentation and justification of the type of quantitative research this is.

d) The first three sections (introduction and the two literature review sections) need further development and support by credible and more recent resources.

Specifically, I would had expected to read more about emergency remote teaching (vs "traditional" distance learning). For example, there is a very similar study which can now be used for the theoretical framework:

https://link.springer.com/article/10.1007/s11618-021-01002-x

The critical literature review of the paper below can be also of interest:

https://online-journals.org/index.php/i-jet/article/view/9406/5446

and of course another excellent literature review section on the same topic:

https://www.ncbi.nlm.nih.gov/pmc/articles/PMC7846229/

e) The reference list doesn't always follow the journal's referencing system

Very strong statistical analysis, I enjoyed reading this part.

---

## [Author Response · Author response to Decision Letter 0]

6 Jul 2021

Protocol of changes and improvements in accordance with the reviewers’ recommendations

To the Editor:

We thank the editor and the reviewers for their feedback and their positive assessment of our manuscript. Your input helped us to further improve our manuscript. 

Below, we addressed each comment by first presenting our answer in detail and furthermore pointing out corresponding changes in the revised manuscript (sections that have been removed are crossed out, whereas parts that have been added are underlined. As requested, we also provide a marked-up revised manuscript as well as the revised manuscript without mark-up. We think that the quality of the manuscript has been significantly improved and wish to thank again the editor and reviewers for the time and attention they devoted to this manuscript. 

To Reviewer 1:

Comments to the Author

1. In one of the initial pages of the material, 17 countries are mentioned, but in reality there are 16.

Data was collected in 17 countries: Albania, Austria, China, Croatia, Estonia, Finland, Germany, Iceland, Japan, Kosovo, Lithuania, Poland, Malta, North Macedonia, Romania, Sweden, and the US. In the abstract (p. 4), only 16 countries were listed, as we forgot to include the project leading country Austria. We have now corrected this mistake. 

Page 4

“A total of N = 15,462 participants from Albania, Austria, China, Croatia, Estonia, Finland, Germany, Iceland, Japan, Kosovo, Lithuania, Poland, Malta, North Macedonia, Romania, Sweden, and the US answered questions regarding perceived competence, autonomy, social relatedness, intrinsic motivation, procrastination, persistence, and sociodemographic background.”

2. Clarify the reason why there are no Latin American countries.

It would have been clearly ideal to collect data from all continents. However, we used existing networks for the recruitment of partner laboratories and researchers. Participation in the project was based on self-selection: Those partners who were interested in cooperation were included in the existing project. In addition, it should be noted that the outbreak of the COVID-19 pandemic in Latin American countries was delayed, and infection levels were low until mid-May (see https://ourworldindata.org/coronavirus#coronavirus-country-profiles). In this respect, they did not fall within the target sample at the time. 

3. Indicate to what the authors attribute the higher number of female vs. male participants.

A more gender balanced sample would have been desirable. However, data were collected completely online, which led to a self-selection of the sample and consequently to an overrepresentation of women; it has been noted by other researchers (Porter & Whitcomb, 2005) that women tend to participate more readily in online studies than men. This common problem in psychological research might be attributed to gender roles promoting a higher willingness to communicate, share personal information or help researchers by participating in a study in women than in men. Unfortunately, we did not have the resources to additionally advertise the study to male participants in particular. 

4. Point out the reason why Austria, Finland and Switzerland have a higher number of participants.

Again, the sample was self-selected, leading to differences in sample size between the participating countries. Each research lab was responsible for collecting the data and recruiting participants. For example, in Austria, the Federal Ministry of Austria for Education, Science and Research supported our study and posted the link to the survey on their website. Moreover, the media have reported about the study. Finland and Sweden (Switzerland wasn’t part of the study) contacted students via faculty email lists and social media (Finland) and emails to universities and student unions (Sweden). More information about the recruitment methods for each respective country can be found in the Online Supplementary Material (pp. 1).

5. Not mentioned in the instrument What were the items added to adapt to the pandemic?

We thank the reviewer for this comment. In the method section (heading “Measures”) we stated that “In order to take the unique situation into account, existing scales were adapted to the current pandemic context and supplemented with a small number of newly developed items” (p. 13). 

More detail is provided in the description of the specific scales, were we state for every scale and item whether or not items were newly constructed (as in the Autonomy scale) or adapted to existing scales (e.g. Perceived competence). Adaptations were mostly made to account for the learning context (e.g. Perceived competence) and/or to the current situation (changing the instructions to “In the current home-learning situation …” or “Currently …”). We have now added this information to the “Measures” section. 

Page 13

“In order to take the unique situation into account, existing scales were adapted to the current pandemic context (e.g., adding “In the current home-learning situation …”), and supplemented with a small number of newly developed items.”

We are happy to provide further information should the reviewer have any more questions. 

To Reviewer 2:

Comments to the Author

6. This is a very interesting, multi-author, multi-country study, which aims to investigate the relationship between basic need satisfaction and procrastination and persistence in the context of emergency distance learning during the COVID-19 pandemic.

This is an interesting paper and the data can be used by other researchers in the field in different countries.

We thank the reviewer for their kind words. 

7. The abstract shouldn't include statistical data and such detail on the participants.

We acknowledge that, depending on the research area, different approaches are taken regarding what should be included in the different parts of the manuscript. We took care to adhere to the Plos One submission guideline, where it is stated that no figures, tables, supporting information or references should be cited in the abstract. We agree that information about the gender distribution may be omitted in the abstract (and therefore deleted it), however we think that providing basic sample descriptives (particularly the sample size) provide additional value for readers who want to determine the relevance of the presented research upon scanning the abstract. This is also in line with recommendations of the American Psychological Association (American Psychological Association, 2020, p. 98) and other recently published articles in PlosOne in the field of psychology (e.g., Ho et al., 2021; Weeldenburg et al., 2020).

Page 4

“A total of N = 15,462 (71.7% female, 27.4% male and 0.7% diverse) participants from Albania, Austria, China, Croatia, Estonia, Finland, Germany, Iceland, Japan, Kosovo, Lithuania, Poland, Malta, North Macedonia, Romania, Sweden, and the US answered questions regarding perceived competence, autonomy, social relatedness, intrinsic motivation, procrastination, persistence, and sociodemographic background.”

8. The "present study" section should be deleted. Important information about the objectives and the methodology of the research can be placed in the introduction and the methods sections respectively.

We thank the reviewer for his comment. We have now changed the manuscript in the following ways:

● We removed the caption “The present study” from the introduction.

● The aim of the study is now described in the introduction, and we explicitly formulated the research questions and hypotheses. 

Formerly Page 9

“The present study

The overall goal of this study was to investigate the well-established relationship between the three basic needs of autonomy, competence and social relatedness with intrinsic motivation in the new and specific situation of pandemic-induced distance learning. We furthermore extend SDT’s predictions regarding two important aspects of learning behavior - procrastination (as a consequence of low or absent intrinsic motivation) and persistence (as the implementation of the volitional part of motivation). Based on SDT's claim of universality, we assume that these relationships will emerge across countries. A multi-country design is particularly interesting in a pandemic setting: During this global health crisis, educational institutions in all countries face the same challenge (to provide distance learning in a way that allows students to succeed) but do so within different frameworks depending on the specific measures each country has implemented. This provides a unique basis for comparing the effects of need fulfillment on students’ learning behavior cross-nationally, thus testing the universality of SDT.

In three different multigroup models, we analyze the relationship between each of the basic needs, persistence, and procrastination, each directly and indirectly mediated by intrinsic motivation. We expect that perceived satisfaction of the basic needs for autonomy (H1a), competence (H1b) and social relatedness (H1c) will be positively related to intrinsic motivation. Intrinsic motivation will be negatively associated with procrastination (H2a - c) but positively associated with persistence (H3a - c). Furthermore, we assume that intrinsic motivation will mediate the relationship between perceived autonomy, competence and social relatedness and procrastination (H4a - c) and persistence (H5a - c). We also propose that perceived autonomy, competence and social relatedness have a direct negative relation with procrastination (H6a - c) and a direct positive relation with persistence (H7a - c). As SDT claims to be a universal theory independent from country and culture, we expect a similar pattern of results in all observed countries (H8a - c). As previous studies have indicated that gender [3,12,33] and age [34,35] may influence intrinsic motivation, persistence and procrastination, we included participants’ gender and age as control variables. A multi-country design is particularly interesting in a pandemic setting: During this global health crisis, educational institutions in all countries face the same challenge (to provide distance learning in a way that allows students to succeed) but do so within different frameworks depending on the specific measures each country has implemented. This provides a unique basis for comparing the effects of need fulfillment on students’ learning behavior cross-nationally, thus testing the universality of SDT.”

Page 9 ff

“Therefore, the overall goal of this study is to investigate the well-established relationship between the three basic needs for autonomy, competence, and social relatedness with intrinsic motivation in the new and specific situation of pandemic-induced distance learning. Firstly, we examine the relationship between each of the basic needs with intrinsic motivation. We expect that perceived satisfaction of the basic needs for autonomy (H1a), competence (H1b) and social relatedness (H1c) would be positively related to intrinsic motivation. In our second research question, we furthermore extend SDT’s predictions regarding two important aspects of learning behavior - procrastination (as a consequence of low or absent intrinsic motivation) and persistence (as the implementation of the volitional part of motivation) and hypothesize that each basic need will be positively related to persistence and negatively related to procrastination, both directly (procrastination: H2a - c; persistence: H3a - c) and mediated by intrinsic motivation (procrastination: H4a - c; persistence: H5a - c). We also proposed that perceived autonomy, competence, and social relatedness would have a direct negative relation with procrastination (H6a - c) and a direct positive relation with persistence (H7a - c). Finally, we investigate SDT's claim of universality, and assume that the aforementioned relationships will emerge across countries we therefore expect a similar pattern of results in all observed countries (H8a - c). As previous studies have indicated that gender (4,17,38) and age (39,40). may influence intrinsic motivation, persistence, and procrastination, we included participants’ gender and age as control variables.”

9. I missed explicit presentation and justification of the type of quantitative research this is.

We thank the reviewer and agree that the explicit mention of the study design was lacking in our manuscript. We added it in the new section “Study design” which is now included in the “Methods” part of the paper. In it, we explain that and why we opted for a cross-sectional multi-country online survey design. We opted for an online-design due to the pandemic-related restrictions on physical contact with potential survey participants as well as due to the potential to reach a larger audience with an online survey. We opted for a cross-sectional rather than a longitudinal design because, firstly, we were more interested in the current state of affairs in schools than in long-term development and, secondly, because we were particularly interested in a large-scale section of the population and longitudinal samples are necessarily smaller. 

We are unsure as to what the reviewer means by “justification” of the type of quantitative research. However, if any further information is needed, we would be happy to add it. 

Page 10

“Study design

Due to the circumstances, we opted for a cross-sectional study design across multiple countries, conducted as an online survey. We decided for an online-design due to the pandemic-related restrictions on physical contact with potential survey participants as well as due to the potential to reach a larger audience. As we were interested in the current situation in schools rather than in long-term development, and we were particularly interested in a large-scale section of the population in multiple countries, we decided on a cross-sectional design. In addition, a multi-country design is particularly interesting in a pandemic setting: During this global health crisis, educational institutions in all countries face the same challenge (to provide distance learning in a way that allows students to succeed) but do so within different frameworks depending on the specific measures each country has implemented. This provides a unique basis for comparing the effects of need fulfillment on students’ learning behavior cross-nationally, thus testing the universality of SDT.”

10. The first three sections (introduction and the two literature review sections) need further development and support by credible and more recent resources.

Specifically, I would had expected to read more about emergency remote teaching (vs "traditional" distance learning). For example, there is a very similar study which can now be used for the theoretical framework:

https://link.springer.com/article/10.1007/s11618-021-01002-x

The critical literature review of the paper below can be also of interest:

https://online-journals.org/index.php/i-jet/article/view/9406/5446

and of course another excellent literature review section on the same topic:

https://www.ncbi.nlm.nih.gov/pmc/articles/PMC7846229/

We thank the reviewer for their comment and the helpful suggested literature. 

We have now further developed the theoretical background in the following ways:

In the “Introduction”, we added a sentence about the difference between planned distance learning and emergency distance learning, to emphasize the need for further research on the topic of pandemic-induced distance learning. 

Page 5

“At universities, instruction was urgently switched to distance learning, bearing challenges for all actors involved, particularly for students (2). Moreover, since distance teaching requires ample preparation time and situation-specific didactic adaptation to be successful, previously established concepts for and research findings on distance learning cannot be applied undifferentiated to the emergency distance learning situation at hand (3). 

Generally, Iit has been shown that the less structured learning environment in distance learning requires students to regulate their learning and motivation more independently (4).”

Furthermore, we included a paragraph in the chapter “The fundamental role of basic psychological needs for intrinsic motivation and learning behavior” where we present previous results on the connection between each basic psychological need and intrinsic motivation. 

Page 6

“SDT proposes that the satisfaction of each of these three basic needs uniquely contributes to intrinsic motivation, a claim that has been proved in numerous studies and in various learning contexts. For example, Martinek and colleagues (10) found that autonomy satisfaction was positively whereas autonomy frustration was negatively related to intrinsic motivation in a sample of university students during COVID-19. The same held true for competence satisfaction and dissatisfaction. A recent study compared secondary school students who perceived themselves as highly competent in dealing with their school-related tasks during pandemic-induced distance learning to those who perceived themselves as low in competence (11). Students with high perceived competence not only reported higher intrinsic motivation but also implemented more self-regulated learning strategies (such as goal setting, planning, time management and metacognitive strategies) and procrastinated less than students who perceived themselves as low in competence. Of the three basic psychological needs, the findings on the influence of social relatedness on intrinsic motivation have been most ambiguous. While in some studies, social relatedness enhanced intrinsic motivation (e.g., (12)), others could not establish a clear connection (e.g., (13)).”

11. The reference list doesn't always follow the journal's referencing system.

We have revised the references and hope that it now complies with the journal’s referencing system.

12. Very strong statistical analysis, I enjoyed reading this part.

We thank the reviewer for the kind comment and hope our changes make the rest of the manuscript equally enjoyable!

References cited in this revision protocol:

American Psychological Association. (2020). Publication manual of the American Psychological Association (7th ed.). https://doi.org/10.1037/0000165-000

Ho, I. M. K., Cheong, K. Y., & Weldon, A. (2021). Predicting student satisfaction of emergency remote learning in higher education during COVID-19 using machine learning techniques. Plos one, 16(4), e0249423. https://doi.org/10.1371/journal.pone.0249423

Porter, S. R., & Whitcomb, M. E. (2005). Non-response in student surveys: The role of demographics, engagement and personality. Research in higher education, 46(2), 127-152. http://doi.org/10.1007/s11162-004-1597-2

Weeldenburg, G., Borghouts, L. B., Slingerland, M., & Vos, S. (2020). Similar but different: Profiling secondary school students based on their perceived motivational climate and psychological need-based experiences in physical education. PloS one, 15(2), e0228859. https://doi.org/10.1371/journal.pone.0228859

---

## [Decision Letter · Decision Letter 1]

31 Aug 2021

Distance learning in higher education during COVID-19: The role of basic psychological needs and intrinsic motivation for persistence and procrastination – A multi-country study

PONE-D-21-10408R1

Dear Dr. Elisabeth Rosa Pelikan,

We’re pleased to inform you that your manuscript has been judged scientifically suitable for publication and will be formally accepted for publication once it meets all outstanding technical requirements.

Kind regards,

Shah Md Atiqul Haq

Academic Editor

PLOS ONE

Additional Editor Comments (optional):

Dear authors,

Congratulations!!!

Your paper has been accepted.

Best regards,

Reviewers' comments:

Reviewer's Responses to Questions

**Comments to the Author**

1. If the authors have adequately addressed your comments raised in a previous round of review and you feel that this manuscript is now acceptable for publication, you may indicate that here to bypass the “Comments to the Author” section, enter your conflict of interest statement in the “Confidential to Editor” section, and submit your "Accept" recommendation.

Reviewer #2: All comments have been addressed

2. Is the manuscript technically sound, and do the data support the conclusions?

Reviewer #2: Yes

3. Has the statistical analysis been performed appropriately and rigorously? 

Reviewer #2: Yes

4. Have the authors made all data underlying the findings in their manuscript fully available?

Reviewer #2: Yes

5. Is the manuscript presented in an intelligible fashion and written in standard English?

Reviewer #2: Yes

6. Review Comments to the Author

Reviewer #2: All my comments and requests have been addressed and I would like to recommend this article for publication.

7. PLOS authors have the option to publish the peer review history of their article (what does this mean?). If published, this will include your full peer review and any attached files.

Reviewer #2: No

---

## [Editor Report · Acceptance letter]

15 Sep 2021

PONE-D-21-10408R1 

Distance learning in higher education during COVID-19: The role of basic psychological needs and intrinsic motivation for persistence and procrastination – A multi-country study 

Dear Dr. Pelikan:

I'm pleased to inform you that your manuscript has been deemed suitable for publication in PLOS ONE. Congratulations! Your manuscript is now with our production department. 

Kind regards, 

on behalf of

Dr. Shah Md Atiqul Haq 

Academic Editor

PLOS ONE